# Findings on emergent magnetic resonance imaging in pregnant patients with suspected appendicitis: A single center perspective

**Hila Bufman**[1,2]*, **Daniel Raskin**[1,2], **Yiftach Barash**[1,2], **Yael Inbar**[1,2], **Roy Mashiach**[2,3], **Noam Tau**[1,2]

**1** Department of Diagnostic Imaging, Sheba Medical Center, Ramat Gan, Israel, **2** Sackler Faculty of Medicine, Tel Aviv University, Tel Aviv, Israel, **3** Department of Obstetrics and Gynecology, Sheba Medical Center, Ramat Gan, Israel

\* HilaBufman@gmail.com

**Data Availability Statement:** Data cannot be shared publicly because of patient confidentiality.

## Abstract

This study's aim is to describe the imaging findings in pregnant patients undergoing emergent MRI for suspected acute appendicitis, and the various alternative diagnoses seen on those MRI scans. This is a single center retrospective analysis in which we assessed the imaging, clinical and pathological data for all consecutive pregnant patients who underwent emergent MRI for suspected acute appendicitis between April 2013 and June 2021. Out of 167 patients, 35 patients (20.9%) were diagnosed with acute appendicitis on MRI. Thirty patients (18%) were diagnosed with an alternative diagnosis on MRI: 17/30 (56.7%) patients had a gynecological source of abdominal pain (e.g. ectopic pregnancy, red degeneration of a leiomyoma); 8 patients (26.7%) had urological findings such as pyelonephritis; and 6 patients (20%) had gastrointestinal diagnoses (e.g. abdominal wall hernia or inflammatory bowel disease). Our conclusions are that MRI is a good diagnostic tool in the pregnant patient, not only in diagnosing acute appendicitis, but also in providing information on alternative diagnoses to acute abdominal pain. Our findings show the various differential diagnoses on emergent MRI in pregnant patients with suspected acute appendicitis, which may assist clinicians and radiologists is patient assessment and imaging utilization.

## Introduction

The diagnosis of abdominal pain in pregnant patients often poses a diagnostic challenge, since clinical assessment is often inconclusive, in part due to normal physiological changes and limited available imaging possibilities. The differential diagnosis for acute abdominal pain in these patients is wide, and can be divided roughly into gynecological (e.g. ovarian torsion and degenerating fibroids), gastrointestinal (e.g., acute appendicitis, bowel obstruction and cholelithiasis) and urological causes (e.g. hydronephrosis, which in 90% of patients is physiological, and pyelonephritis) [1–3]. Acute appendicitis during pregnancy is the second most common indication for surgery in pregnant patients after cesarean section, with a prevalence of approximately one in 500–650 pregnancies per year [4, 5]. The risk of misdiagnosing acute appendicitis in these patients may lead to a grave prognosis compared to non-pregnant patients [6, 7]

Data are available from the Sheba Medical Center Institutional Data Access / Ethics Committee (contact via phone number: +97235305997) for researchers who meet the criteria for access to confidential data.

**Funding:** The authors received no funding for this study.

**Competing interests:** The authors have declared no competing interest exist.

and is related to fetal loss [8]. Given the changes in anatomy during the later stages of pregnancy, and other potential causes of acute abdominal pain, accurate diagnosis of acute appendicitis poses a challenging diagnostic dilemma. Therefore, there is a need for an accurate and safe imaging modality for these patients.

When a pregnant patient presents with acute abdominal pain suspected for acute appendicitis, the first-line imaging modality is ultrasonography (US) [9, 10]. However, although US is a safe and readily available imaging modality, it has low sensitivity ranging from 12–27% and may be inconclusive or fail to demonstrate the appendix altogether, with identification of the appendix in as low as 2% of patients [10–13]. Use of Computed Tomography is not considered an option in this setting due to the exposure to ionizing radiation, not recommended during pregnancy [11].

Magnetic Resonance Imaging (MRI) was shown to have a good diagnostic value with sensitivity and specificity of over 90% in most studies [13–15], not only in visualizing the appendix, but also in diagnosing alternative causes of acute abdominal pain, including pyelonephritis and cholecystitis [16, 17]. According to the American College of Gynecology (ACOG), Non-contrast MRI is safe to use in pregnant patients in all trimesters, including first trimester. Gadolinium is water soluble and may cross the placenta to the amniotic fluid and was found to be teratogenic in animal studies thus it is contraindicated in the pregnant patients [11]. Various studies proved MRI to be superior to US in these patients [13, 18]. To the best of our knowledge, there are no studies describing the prevalence of various pathological imaging findings on emergent MRI performed in pregnant patients for suspected acute appendicitis.

The aim of our study was to assess and describe the imaging findings on emergent MRI performed to rule out acute appendicitis in pregnant patients presenting to the emergency department (ED), and show the various diagnoses found on MRI in these patients.

## Methods

This was an institutional review board approved retrospective study conducted in a single tertiary academic medical center. Informed consent was waived as this is a retrospective study and data was analyzed anonymously. In our study, we included all consecutive pregnant patients who underwent emergent non-contrast enhanced abdominal-pelvic MRI for suspected acute appendicitis between April 1, 2013 and June 31, 2021. All included patients presented to our emergency department (ED) with right lower quadrant pain and underwent medical and surgical assessment which raised the suspicion of acute appendicitis. We included all adult pregnant patients (>18 years old) at the time of ED admission underwent emergent non-contrast enhanced abdominal-pelvic MRI for suspected acute appendicitis. We excluded patients younger than 18 years old or for whom there was no complete medical data in our institutional electronic patient file. For each patient we collected the following data: Patient age; gestational week; number of fetuses; gravidity number; and parity number.

All patients underwent gynecological US prior to MRI and were only referred to further imaging if gynecological US did not find a cause of abdominal pain.

Acute appendicitis was diagnosed on US according to accepted diagnostic criteria: width >6 mm; lack of compressibility; and inflamed echogenic peri-appendiceal fat. Indeterminate appendicitis was defined as one or more of the following: inflamed echogenic peri-appendiceal fat; right lower quadrant fluid collection; or enlarged mesenterial lymph nodes without the visualization of a distended and non-compressible appendix [19, 20]. Our non-contrast enhanced MRI protocol for suspected appendicitis during pregnancy is detailed in Table 1. All patients received 1 liter of 5% Mannitol as an MRI-positive oral contrast 2 hours before the MRI was performed, with no overall major side-effects.

**Table 1. MRI protocol used.**

| Series | TR (ms) | TE (ms) | Slice thickness (mm) | Gap (mm) | Matrix size |
|---|---|---|---|---|---|
| Coronal T2 SSH[e] | 1250 | 80 | 5 | 0.5 | 416*336 |
| Coronal T2 SPAIR SSH | 1250 | 80 | 5 | 0.5 | 416*336 |
| Coronal BTFE [a] | 3.7 | 1.86 | 5 | 0.5 | 332*264 |
| Coronal BTFE cine | 3.7 | 1.86 | 6 | 1 | 236*161 |
| Axial T2 | 1059 | 80 | 5 | 0.5 | 308*185 |
| Axial T2 + FS [c] | 1131 | 80 | 5 | 0.5 | 308*185 |
| Sagittal T2 | Shortest | 80 | 5 | 0.5 | 368*216 |
| Coronal eTHRIVE | 3.8 | 1.85 | 6 | -3 | 456*404 |
| Sagittal T2 SPAIR | 1250 | 80 | 5 | 0.5 | 368*217 |
| Axial eTHRIVE [b] | 2.8 | 1.3 | 5 | -2.5 | 156*196 |
| 3D Free Breathing | Shortest | shortest | 3 | 0 | 456*404 |

[a]BTFE–Balanced turbo field echo,

[b]eTHRIVE–enhanced T1 high resolution isotropic volume excitation,

[c]FS–Fat suppression,

[e]SSH–single-shot turbo spin echo.

The MRI images of all included patients were re-assessed by a 4[th] year radiology resident, with oversight by a fellowship trained abdominal radiologist with 11 years of experience, in case of disagreement regarding the diagnosis between the two radiologists, the case was reevaluated and the more experienced radiologist's opinion was the final decision. For each MRI, the following data was collected: Whether the appendix could be identified; whether there were evidence of acute appendicitis (appendix measuring > 7 mm in outer diameter and/or periappendiceal inflammation or fluid collection [21]); and the existence of an alternative diagnosis, if any. Post-MRI treatment data was collected from the patient file (including surgical, pathological and medical treatment). Final diagnosis was determined according to pathological reports if patient underwent surgery, or clinical improvement if patient was managed only medically.

Continuous parameters are presented as median (interquartile range (IQR)). Statistical analysis was performed using SPSS Statistics software (version 27.0, IBM), proportions were described as percentage. Mean and median were calculated. Kappa rate was calculated according to Cohen's Kappa, and is presented with confidence interval (CI).

## Results

Overall, between April 1, 2013 and June 31, 2021, 177 consecutive pregnant patients underwent emergent non-contrast abdominal-pelvic MRI for suspected appendicitis. Three of these patients were younger than 18 years old and 7 other patients did not have available medical files and were excluded from the study.

There were five cases of disagreement between readers for which the final categorization was assigned according to the senior radiologist's reading. Cohen's Kappa for inter-reader agreement was 0.9 (CI 0.818–0.916).

Therefore, our final cohort consisted of 167 female pregnant patients. The median age was 30 years (IQR, 27–34) and the median gestational week was 20 weeks (IQR,15–26) with a number of fetuses of 1. The median gravidity number was 2 pregnancies (IQR, 1–3), and the

median parity number was 1 prior birth (IQR, 0–1). Patient demographics are presented in Table 2.

Most patients underwent ultrasound examination prior to MRI (153/167; 91.6%). Of these, 110/153 (71.9%) had a normal US examination. Definite appendicitis was diagnosed in 4 patients (2.6%), and US was deemed indeterminate in 23/153 (15%) patients. Non appendiceal sources of abdominal pain are detailed in Table 3. Of the patients with a normal US examination, 23/110 (20.9%) patients were later diagnosed with a pathology on MRI: 17 had acute appendicitis and 6 had various non-appendiceal pathologies.

On MRI, the appendix was identified in 140/167 (83.8%) patients. Acute appendicitis was diagnosed in 35/167 (20.9%) patients (Fig 1), and 30/167 (18%) patients were found to have an alternative diagnosis, as detailed in Table 4 and Fig 2.

Gynecological sources of acute pain were found in 16/167 (9.5%) patients included ectopic pregnancy, red degeneration of a leiomyoma and others. Urological pathologies mostly included pyelonephritis, as seen in 8/167 (4.8%) patients (Figs 3 and 4). Surgical diagnoses included, among others, rectus sheath hematoma (Fig 5) and IBD-related pathologies. In our cohort, 102/167 (61%) patients did not have any acute findings on MRI to account for the presenting abdominal pain.

Overall, 33 (19.7%) patients received antibiotic treatment, either for a known pathology such as pyelonephritis, or as an empirical treatment without proven source of infection.

Thirty-three other patients (19.7%) underwent surgery, in whom the majority (30/33; 90.9%) was appendectomy, and among those, acute appendicitis was corroborated on pathology in most (27/30; 90%). Further two patients underwent emergent salpingectomy for ectopic pregnancy and one patient underwent explorative laparoscopy to rule out ovarian torsion.

Of the 61% (102/167) of patients who had a negative MRI, one was diagnosed with amnionitis (1/102, 0.9%), one with a miscarriage (1/102, 0.9%), one with cystitis (1/102, 0.9%) and one with ovarian vein thrombosis (1/102, 0.9%).

As most obstetric patients in our healthcare system are not followed in hospital but rather in community-based clinics not affiliated to our hospital, we do not have access to long term follow up for most patients. However, as delivery data is available to us, we have found that

**Table 2. Patient demographics.**

| Total patients | 167 |
|---|---|
| Age (years) | |
| Mean | 30.4 |
| Median | 30 |
| Range | 19–46 |
| gestational age | |
| Mean | 19.9 |
| Median | 20 |
| Range | 3–37 |
| Gravidity | |
| Mean | 2.6 |
| Median | 2 |
| Range | 1–14 |
| Parity | |
| Mean | 1.1 |
| Median | 1 |
| Range | 0–10 |

**Table 3. Ultrasound findings prior to MRI.**

| Finding | Patients (n = 167) |
|---|---|
| US [a] was performed | 153 (91.6) |
| Appendix was not visualized and there were no other findings to account for abdominal pain. | 110 (71.9) |
| Definite appendicitis | 4 (2.6) |
| Normal appendix identified | 1 (0.6) |
| Indeterminate appendicitis | 23 (15) |
| Gallstones | 3 (1.9) |
| Hydronephrosis | 7 (4.5) |
| Gynecological source of abdominal pain | 4 (2.6) |
| Surgical source of abdominal pain | 1 (0.6) |

Note—Values expressed as number (percentage)

[a]–US—Ultrasound

none of the patients who presented to hospital with RLQ pain and did not have acute appendicitis had a preterm delivery or any other obstetric complications, regardless of whether the MRI found a cause for their pain or not.

## Discussion

As the use of emergent MRI scans increased in recent years [22], there is a need to better describe the findings found on those scans, especially in pregnant patients undergoing MRI in acute abdominal pain.

(a)

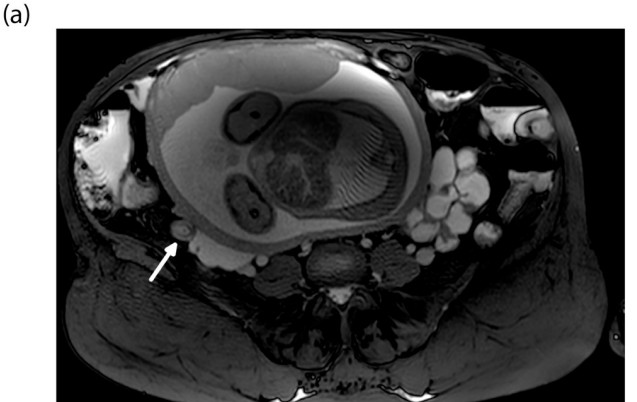

(b)

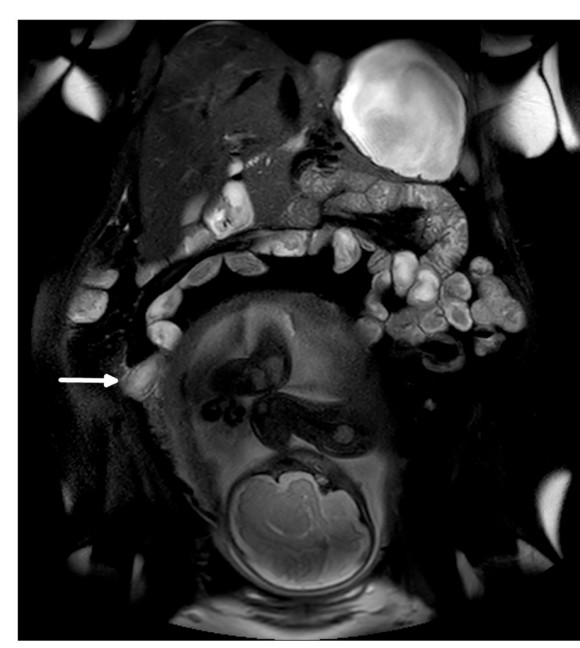

**Fig 1. Magnetic resonance imaging of a 36-year-old, 32-week pregnant patient who presented with right lower quadrant pain.** A and B, Axial (A) and Coronal (B) fat saturated T2 weighted images showing a distended retrocecal appendix (arrow), a position which makes it difficult to demonstrate on ultrasonography. These findings are consistent with acute appendicitis, which was also confirmed on surgery and pathology.

**Table 4. Pathologies found on MRI.**

| Group | Finding | Number of patients |
|---|---|---|
| Gynecological | | 16 / 167 (9.5) |
| | Leiomyoma | 6 / 16 (37.5) |
| | Hemorrhagic Corpus Luteum | 3 / 16 (18.7) |
| | Ectopic Pregnancy | 2 / 16 (12.5) |
| | Endometriosis | 2 / 16 (12.5) |
| | Placental Hematoma | 2 / 16 (12.5) |
| | Ovarian Hyperstimulation Syndrome | 1 /16 (6.3) |
| Urological | | 8 / 167 (4.7) |
| | Pyelonephritis | 3 / 8(37.5) |
| | Hydronephrosis | 3 / 8 (62.5) |
| Surgical | | 6 / 167 (3.6) |
| | Post Appendectomy Fluid Collection | 2 / 6 (33.3) |
| | Incarcerated Hernia | 2 / 6 (33.3) |
| | Rectus Sheath Hematoma | 1 / 6 (16.7) |
| | Crohn's Exacerbation | 1 / 6 (16.7) |

Note—Values expressed as number (percentage)

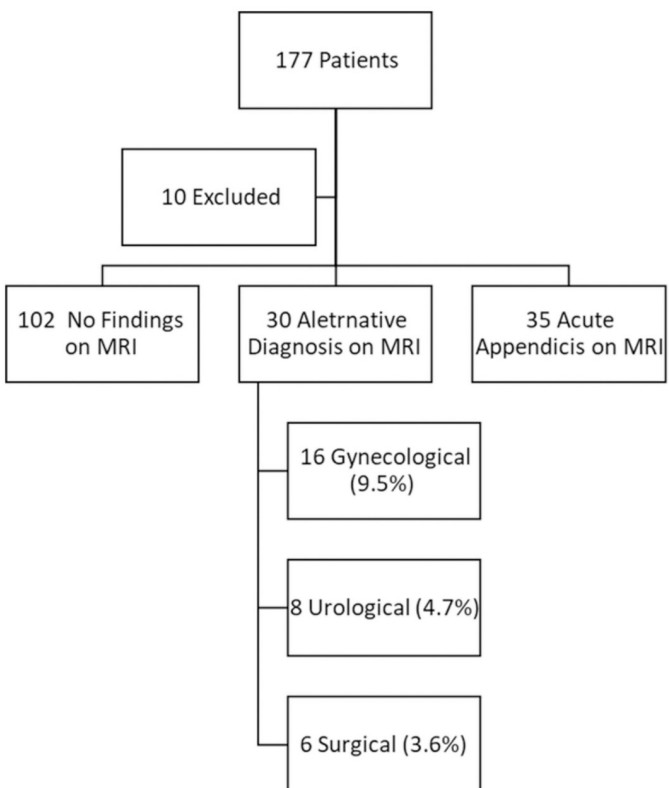

**Fig 2. Flow chart depicting MRI diagnosis.**

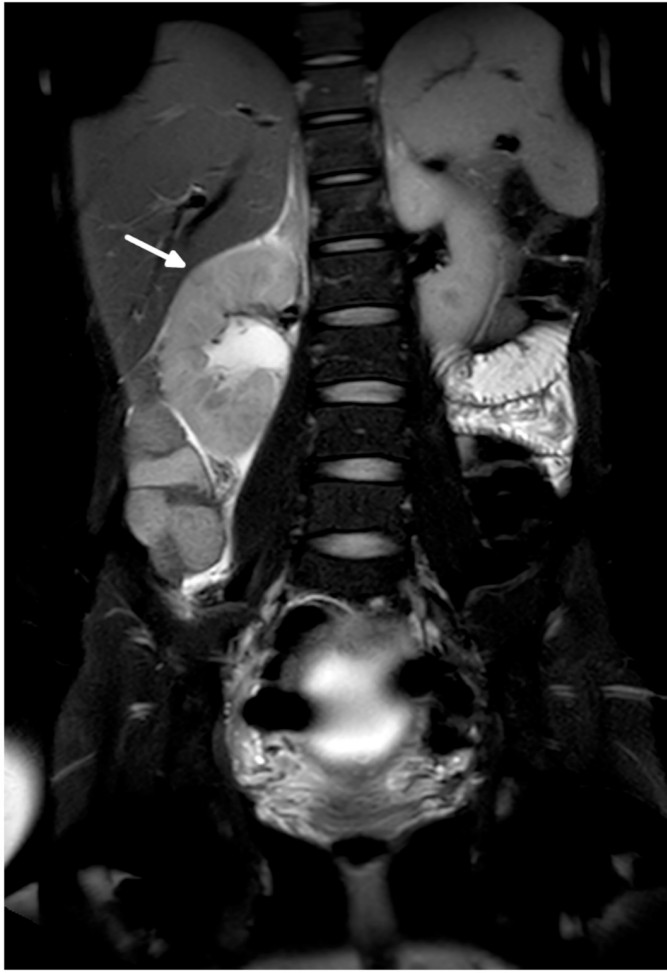

**Fig 3. Coronal fat saturated T2 weighted image of a 30-year-old, 17-week pregnant patient.** There is a right-sided mild to moderate hydronephrosis with regions of parenchymal hypointensity (arrow), as well as peri-nephric fluid and fat stranding, all consistent with acute pyelonephritis.

In our study, acute appendicitis was diagnosed in 20.9% of pregnant patients presenting to the ED with acute right lower quadrant pain. Most of the patients who underwent surgery had an appendectomy, with a correct imaging and pathological correlation rate of 90%, similar to results in previous studies [10, 23].

Our study also describes alternative causes for acute right lower quadrant pain identified on MRI scans performed to rule out acute appendicitis. These alternative pathologies were found almost as often as acute appendicitis (18% vs. 20.9%, respectively). The main differential diagnoses identified in our study were of gynecological (in 10.2% of patients) as well as surgical and urological, unexpected given that the patients have undergone a thorough clinical assessment prior to MRI, including a gynecological examination and laboratory tests. To the best of our knowledge, there are no other studies describing the frequency of those alternative diagnoses in this patient group undergoing emergent MRI.

Our findings not only confirm the known capability of MRI in correctly diagnosing acute appendicitis in pregnant patients, but also add other possible diagnoses in this patient group and can serve as an important addition to the growing body of evidence supporting the use of

(a)

(b)

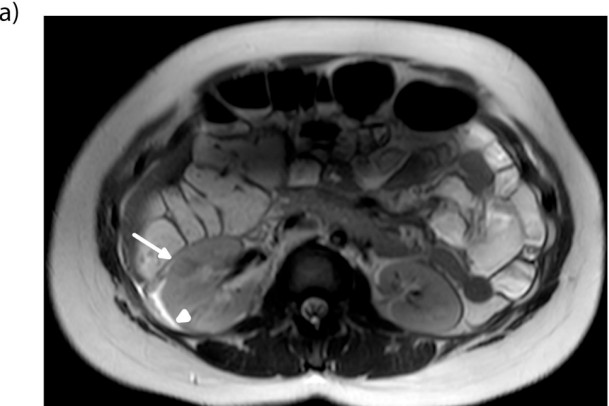
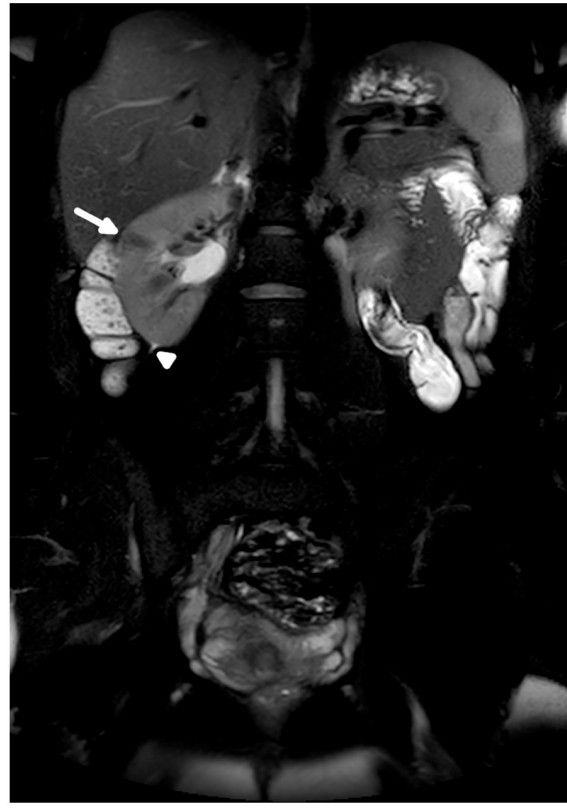

**Fig 4. A 26-week pregnant, 34-year-old patient who presented with right lower quadrant pain.** Axial T2 weighted image (A) shows a swollen right kidney with marked hypointense lesions in the parenchyma (arrow) and a small amount of peri-nephric free fluid (arrowhead). Coronal fat saturated T2 weighted image (B) shows hypointense lesions in the right kidney's parenchyma (arrow) and a small amount of peri-nephric free fluid (arrowhead). Findings are consistent with acute pyelonephritis.

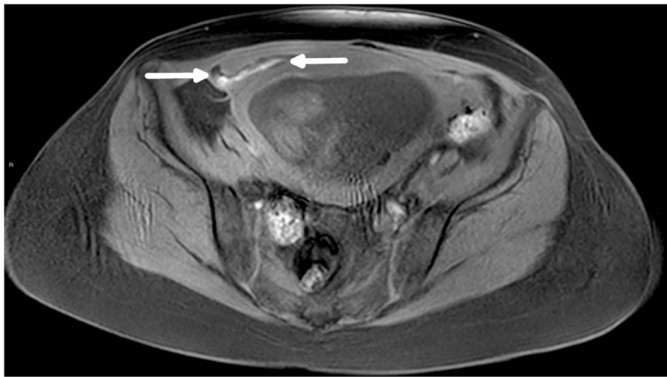

**Fig 5. 41-year-old patient, 19 weeks pregnant, who presented with right lower quadrant pain.** Axial fat saturated T1 weighted image shows hyperintense fluid at the posterior border of the right rectus abdominis muscle (arrows). This finding is consistent with rectus sheath hematoma.

MRI in emergent patient imaging. The diagnostic capabilities of MRI, and its capability to rule out a major abdominal pathology in patients with a negative scan, can likely benefit clinicians in the management of pregnant patients with right lower quadrant pain, especially when no findings on clinical examination can account for the source of the pain. Knowledge of these specific alternative diagnoses may help clinicians to use proper consultations and correctly refer the patient for further work-up, if needed. Future studies need to be performed to assess whether performing emergent MRI can lead to shorter hospitalization periods and a decrease in use of empirical antibiotic treatment after a normal MRI scan. It should also be noted that in our patient cohort, more than 60% of patients had no pathology on MRI, which likely represents the relative ease of using MRI in this patient population at our clinical setting, and the fact that clinicians opt to utilize MRI when face with diagnostic doubt. This high proportion of non-pathologic scans was deemed acceptable by internal discussions in our institute, given the severe potential complications of missing acute appendicitis or other important pathologies in pregnant patients.

Although MRI is not yet widely available for the use of diagnosis of right lower quadrant pain in the pregnant patients, our research supports findings shown in prior studies as to the superiority of MRI over US [13, 18], not only in diagnosing acute appendicitis, but also in diagnosing alternative diagnoses not suspected after initial US examinations. These important diagnoses might have been missed without the use of emergent MRI, leading to the possibility of expanding the use of emergent MRI in pregnant patients with generalized abdominal pain, rather than only in those with suspected acute appendicitis.

Although our study did not focus on US in the settings of acute RLQ pain in the pregnant patient, it is noticeable that alternative diagnoses proposed on MRI were not diagnosed on US, such as ruptured ectopic pregnancy or hematosaplinx, pyelonephritis or incarcerated umbilical hernia. As US in the pregnant patient has limitations due to patient habitus and physiological changes, and as MRI is readily available in our intuition for these patients, final patient management was determined according to a combination of clinical and MRI findings in all cases. This can emphasize the usefulness of MRI in pregnant patients with RLQ pain, compared to using US only.

This study has some limitations. By the nature its retrospective design, the clinical correlation was based on medical patient files, which may be lacking or inaccurate. Second, as patients were assessed by various surgeons and ED physicians prior to MRI, the degree of suspicion of acute appendicitis may have varied. However, as all included patients underwent MRI, we assume that the degree of suspicion and pre-test probability for acute appendicitis was high enough to warrant such a scan. Finally, this was a single-center study, with a dedicated MRI protocol performed by MRI technologists trained to image pregnant patients. Such facilities and protocols may not be available in every hospital, which may limit the generalizability of our findings.

In conclusion, our study showed that the use of emergent MRI in pregnant patients with acute abdominal pain can serve as a good diagnostic tool both for acute appendicitis and in finding alternative diagnoses. Future large scale prospective studies may help in better identifying those patients who may benefit most from undergoing emergent MRI in this clinical setting.

## Author Contributions

**Conceptualization:** Daniel Raskin, Noam Tau.

**Data curation:** Hila Bufman, Yiftach Barash.

**Formal analysis:** Hila Bufman.

**Methodology:** Noam Tau.

**Supervision:** Noam Tau.

**Validation:** Noam Tau.

**Visualization:** Noam Tau.

**Writing – original draft:** Hila Bufman.

**Writing – review & editing:** Hila Bufman, Yael Inbar, Roy Mashiach, Noam Tau.

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
