## [Decision Letter · Decision Letter 0]

6 Mar 2023

PONE-D-23-00975Findings on Emergent Magnetic Resonance Imaging in Pregnant Patients with Suspected Appendicitis: A Single Center PerspectivePLOS ONE

Dear Dr. Bufman,

Thank you for submitting your manuscript to PLOS ONE. After careful consideration, we feel that it has merit but does not fully meet PLOS ONE’s publication criteria as it currently stands. Therefore, we invite you to submit a revised version of the manuscript that addresses the points raised during the review process.

We look forward to receiving your revised manuscript.

Kind regards,

Marc Reismann, MD, PhD

Academic Editor

PLOS ONE

Journal Requirements:

"NO"

"NO"

Additional Editor Comments:

The reviewers focus primarily on the lack of particular data and the way of presentation. Thus, both point at the significance of tables with patient demographics. Technical data like scan parameters should be demonstrated in more detail. Alternative methods like ultrasound – and respective findings, if present – should be presented and discussed in more detail. The significance of the presented study against the background of already existing investigations should be worked out. The individual comments of the reviewers can be found below.

Reviewers' comments:

Reviewer's Responses to Questions

**Comments to the Author**

1. Is the manuscript technically sound, and do the data support the conclusions?

Reviewer #1: Partly

Reviewer #2: No

2. Has the statistical analysis been performed appropriately and rigorously? 

Reviewer #1: No

Reviewer #2: N/A

3. Have the authors made all data underlying the findings in their manuscript fully available?

Reviewer #1: No

Reviewer #2: No

4. Is the manuscript presented in an intelligible fashion and written in standard English?

Reviewer #1: Yes

Reviewer #2: Yes

5. Review Comments to the Author

Reviewer #1: I'm pleased to review an original article entitled “Findings on Emergent Magnetic Resonance Imaging in Pregnant Patients with Suspected Appendicitis: A Single Center Perspective”. This topic is clinically significant and the presented images in the Figures are great; however, there are critical limitations in the study design and methodology.

Major comments

The scan parameters in the MR protocol are insufficient (i.e., lack of data except for TR and TE). The parameters must be more detailed to allow other investigators to reproduce them.

The formal reading session with an independent reading by subspecialized GI or ER radiologists is preferable for this study design. The design of this study does not show inter-reader disagreement and differences between radiologists with different experience years.

It is unknown how the final diagnosis was determined. The authors would clarify how many specific diagnoses were made and which evidence (surgery, pathology, clinical course, etc…)

Minor comments

“gastrointestinal” is better than “surgical” (P4L5)

Oral contrast administration is risky for GI symptoms even if the patients were clinically suspected of acute appendicitis. Some patients with bowel obstruction complain of right lower abdominal pain. (P4L14)

Please describe more details for the statistical approach.

There is no definition for definite and indeterminate acute appendicitis in US.

Results

No 95% IC for sensitivity and specificity.

The patient demographics are recommended to be summarized in a Table.

Discussion

The authors could discuss the results in other aspects (e.g., the discrepancy between the results on US and MR in each category [gastrointestinal, gynecological, and urological disease]).

Table

It is unclear the difference between “no pathological findings” and “normal appendix identified” in Table 2.

Reviewer #2: This is a case series on finding from emergent magnetic resonance imaging in pregnant women with suspected appendicitis. The study description is primarily based on the MR findings.

There is no table with description of the population included in the study and the clinical investigations and characteristics. A figure with a flow diagram would be useful.

Where initial gynecological ultrasound performed? It is rather surprising an ectopic pregnancy was diagnosed by MR.

The manus would benefit from a rewriting with clinical perspective and MR findings.

6. PLOS authors have the option to publish the peer review history of their article (what does this mean?). If published, this will include your full peer review and any attached files.

Reviewer #1: No

Reviewer #2: No

---

## [Author Response · Author response to Decision Letter 0]

18 Apr 2023

Dear Doctor Reisman, 

Please find below a point by point response to the reviewers' comments.

The reviewers’ comments greatly improved our study and we hope that it is now fit for publication in you esteemed journal.

 

Reviewer #1:

Major comments

The scan parameters in the MR protocol are insufficient (i.e., lack of data except for TR and TE). The parameters must be more detailed to allow other investigators to reproduce them.

As suggested, we've added the following parameters to the MRI protocol table (Table 1): slice thickness, gap and matrix.

The formal reading session with an independent reading by subspecialized GI or ER radiologists is preferable for this study design. The design of this study does not show inter-reader disagreement and differences between radiologists with different experience years.

As suggested, we've added to the results a kappa value in order to clarify inter-reader agreement: " There were five cases of disagreement between readers for which the final categorization was assigned according to the senior radiologist’s reading. Cohen’s Kappa for inter-reader agreement was 0.9 (CI 0.818-0.916)."

It is unknown how the final diagnosis was determined. The authors would clarify how many specific diagnoses were made and which evidence (surgery, pathology, clinical course, etc…)

Thank you for your comment. We have added a clarification to the Methods section regarding final diagnosis: "Final diagnosis was determined according to pathological reports if patient underwent surgery, or clinical improvement if patient was managed only medically."

Minor comments

“gastrointestinal” is better than “surgical” (P4L5)

This was changed according to your suggestion.

Oral contrast administration is risky for GI symptoms even if the patients were clinically suspected of acute appendicitis. Some patients with bowel obstruction complain of right lower abdominal pain. (P4L14)

Thank you for this important comment. 

As mannitol is a class C contrast agent during pregnancy [1], there is no evidence of any adverse fetal effects. As to possible side effects, in a study evaluating side effects caused by mannitol digestion, the severity of side effects was related to the amount of mannitol administered. With an amount of 1000 ml (similar to that used in our study), gastrointestinal side effects were uncommon [2].

[1] Product Information: ARIDOL(TM) inhalation powder, mannitol inhalation powder. Pharmaxis, Inc, Exton, PA, 2010.

[2] Ajaj, W., Goehde, S.C., Schneemann, H., Ruehm, S.G., Debatin, J.F. and Lauenstein, T.C. (2004), Dose optimization of mannitol solution for small bowel distension in MRI. J. Magn. Reson. Imaging, 20: 648-653.

Please describe more details for the statistical approach.

The statistical approach was described in further detail in the Methods section:. " Continuous parameters are presented as median (interquartile range (IQR)). Statistical analysis was performed using SPSS Statistics software (version 27.0, IBM), proportions were described as percentage. Mean and median were calculated. Kappa rate was calculated according to Cohen's Kappa, and is presented with confidence interval (CI)."

There is no definition for definite and indeterminate acute appendicitis in US.

Definite appendicitis was diagnosed as accepted in radiological literature [3,4] as appendicular width > 6 mm, lack of compressibility, inflamed echogenic periappendiceal fat. Indeterminate appendicitis was defined as one or more of the following: inflamed echogenic periappendiceal fat, right lower quadrant fluid collection or enlarged mesenterial lymph nodes without the visualization of a widened and non-compressible appendix. This was also added to the Methods section.

[3] Yu SH, Kim CB, Park JW, Kim MS, Radosevich DM. Ultrasonography in the Diagnosis of Appendicitis: Evaluation by Meta-analysis. Korean J Radiol. 2005 Oct-Dec;6(4):267-277.

[4] Barbara Hertzberg, William Middleton. Ultrasound: The Requisites, 3rd Edition - July 13, 2015.

Results

No 95% IC for sensitivity and specificity.

As the examples of sensitivity and specificity were derived from the literature which did not contain CIs, we could not provide this data. 

The patient demographics are recommended to be summarized in a Table.

As suggested, table 2 was added to the manuscript and contains patient demographics.

Discussion

The authors could discuss the results in other aspects (e.g., the discrepancy between the results on US and MR in each category [gastrointestinal, gynecological, and urological disease]).

Thank you for this comment. We added to the Discussion section the following paragraph: “Although our study did not focus on US in the settings of acute RLQ pain in the pregnant patient, it is noticeable that alternative diagnoses proposed on MRI were not diagnosed on US, such as ruptured ectopic pregnancy or hematosaplinx, pyelonephritis or incarcerated umbilical hernia. As US in the pregnant patient has limitations due to patient habitus and physiological changes, and as MRI is readily available in our intuition for these patients, final patient management was determined according to a combination of clinical and MRI findings in all cases. This can emphasize the usefulness of MRI in pregnant patients with RLQ pain, compared to using US only. “

Table

It is unclear the difference between “no pathological findings” and “normal appendix identified” in Table 2.

We changed the term "no pathological findings" to "Appendix was not visualized and there were no other findings to account for abdominal pain.”

 

Reviewer #2:

 There is no table with description of the population included in the study and the clinical investigations and characteristics. A figure with a flow diagram would be useful.

Please find our response to reviewer 1, comment 9 (new Table 2). We have also added a flow chart as figure 1. 

Were initial gynecological ultrasound performed? It is rather surprising an ectopic pregnancy was diagnosed by MR.

Yes, all patients underwent gynecological US prior to MRI. We added a clarification in the Methods section: "All patients underwent gynecological US prior to MRI and were only referred to further imaging if gynecological US did not find a cause of abdominal pain. "

The manus would benefit from a rewriting with clinical perspective and MR findings.

As suggested, the paper was reviewed and some clinical aspects were added.

---

## [Decision Letter · Decision Letter 1]

24 May 2023

PONE-D-23-00975R1

Findings on Emergent Magnetic Resonance Imaging in Pregnant Patients with Suspected Appendicitis: A Single Center Perspective

PLOS ONE

Dear Dr. Bufman,

Thank you for submitting your manuscript to PLOS ONE. After careful consideration, we feel that it has merit but does not fully meet PLOS ONE’s publication criteria as it currently stands. Therefore, we invite you to submit a revised version of the manuscript that addresses the points raised during the review process.

Previous reviewer remarks have been satisfactorily answered. There are some minor concerns left within the current review process. These concern primarily clinical procedural issues, specific points regarding MRI in pregnancy and outcome questions. I would appreciate if you could clarify these issues based on the reviewers´ remarks to be found below.

We look forward to receiving your revised manuscript.

Kind regards,

Marc Reismann, MD, PhD

Academic Editor

PLOS ONE

Journal Requirements:

Reviewers' comments:

Reviewer's Responses to Questions

**Comments to the Author**

1. If the authors have adequately addressed your comments raised in a previous round of review and you feel that this manuscript is now acceptable for publication, you may indicate that here to bypass the “Comments to the Author” section, enter your conflict of interest statement in the “Confidential to Editor” section, and submit your "Accept" recommendation.

Reviewer #2: All comments have been addressed

Reviewer #3: (No Response)

2. Is the manuscript technically sound, and do the data support the conclusions?

Reviewer #2: Yes

Reviewer #3: Yes

3. Has the statistical analysis been performed appropriately and rigorously? 

Reviewer #2: Yes

Reviewer #3: Yes

4. Have the authors made all data underlying the findings in their manuscript fully available?

Reviewer #2: No

Reviewer #3: Yes

5. Is the manuscript presented in an intelligible fashion and written in standard English?

Reviewer #2: Yes

Reviewer #3: Yes

6. Review Comments to the Author

Reviewer #2: The manuscript has been revised according to the reviewers comments within the possibility within the studydesign and the data.

Reviewer #3: Dear editor, dear authors,

Thank you for the opportunity to review the paper „Findings on emergent magnetic resonance imaging in pregnant patients with suspected appendicitis: a single center experience“. The paper deals with an important topic- the imaging of suspected appendicitis in pregnant patients. The authors present a consecutive series of 167 patients with this clinical scenario, who underwent MRI. In their cohort around 21% of patients with clinically suspected appendicitis showed appendicitis on MRI and underwent surgery. 18% showed another pathology on MRI and in 60% no imaging diagnosis explaining the clinical picture could be made on MRI. This large series gives important insights in the frequency of possible diagnosis and the results should therefore be published. The paper is written well and there are high quality figures showing the imaging findings.

But there are a small number of revisions the authors should made before he paper can be accepted for publication:

- In most institutions there is always a shortage of MRI capacity. The authors present a rather complex MR program. How long is the examination time? Are there essential sequences allowing a faster examination? Is there a 10 Minute examination possible?

- The authors should comment on the safety of MRI for pregnant patient and the baby for the non radiologist reader. And they should comment on the contraindication of gadolinium based contrast media especially in the first trimester.

- A crucial question is the outcome of the patients without a diagnosis on MRI. How was the patient outcome assessed? Has there been a systematic chart review? It would be interesting how many of the patients with negative MRI showed cystitis, a diagnosis usually not detectable by MRI. A more systematic report of the patients outcome would greatly enhance the paper.

Best regards

7. PLOS authors have the option to publish the peer review history of their article (what does this mean?). If published, this will include your full peer review and any attached files.

Reviewer #2: No

Reviewer #3: No

---

## [Author Response · Author response to Decision Letter 1]

2 Jun 2023

Dear Dr. Reisman, 

Thank you for reviewing our paper.

Please find below a point by point response to the reviewer's comments.

The reviewer's comments greatly improved our study and we hope that it is now fit for publication in you esteemed journal.

Reviewer #3:

In most institutions there is always a shortage of MRI capacity. The authors present a rather complex MR program. How long is the examination time? Are there essential sequences allowing a faster examination? Is there a 10 Minute examination possible?

Our medical center is a large tertiary center, the largest in the Middle East, and we have the privilege of having multiple MRI machines available, and therefore we are always able to accommodate emergent MRI scans whenever required, around the clock. As to a possible abbreviated MRI protocol, our study did not focus on optimizing the complex protocol and are unable to assess whether a shorter protocol is feasible. In our clinical experience of over a decade of emergent MR imaging for possible acute appendicitis in pregnant patients, there is no single sequence which can serve as an optimal sequence to visualize the appendix, and in every patient the appendix is best visualized in different sequences. For that reason, and the readily available MRI time, we do decided not use an abbreviated MRI protocol, and our protocol takes 28 minutes to perform. Further future studies are required to validate such abbreviated MRI protocol. 

The authors should comment on the safety of MRI for pregnant patient and the baby for the non radiologist reader. And they should comment on the contraindication of gadolinium based contrast media especially in the first trimester.

Thank for this important comment. We have added the following explanation in the introduction section: " According to the American College of Gynecology (ACOG), Non-contrast MRI is safe to use in pregnant patients in all trimesters, including first trimester. Gadolinium is water soluble and may cross the placenta to the amniotic fluid and was found to be teratogenic in animal studies thus it is contraindicated in the pregnant patients." 

A crucial question is the outcome of the patients without a diagnosis on MRI. How was the patient outcome assessed? Has there been a systematic chart review? It would be interesting how many of the patients with negative MRI showed cystitis, a diagnosis usually not detectable by MRI. A more systematic report of the patients outcome would greatly enhance the paper.

Most patients who did not have a clear diagnosis on MRI were admitted to the high risk pregnancy unit and were discharged after a clinical improvement, some without any specific treatment and some with the use of empiric antibiotics. Given the nature of our local health system, electronic patient files for community follow-up are not available to us, and in most cases we indeed did not have a long period follow up. The few patients who were followed in our hospital clinics had a long follow up available to us. Overall, only one patient was discharged with the diagnosis of cystitis, not seen on MRI. We have added to the results section the data related to UTI and the following paragraph regarding long term follow up: "As most obstetric patients in our healthcare system are not followed in hospital but rather in community-based clinics not affiliated to our hospital, we do not have access to long term follow up for most patients. However, as delivery data is available to us, we have found that none of the patients who presented to hospital with RLQ pain and did not have acute appendicitis had a preterm delivery or any other obstetric complications, regardless of whether the MRI found a cause for their pain or not."

---

## [Decision Letter · Decision Letter 2]

20 Jun 2023

Findings on Emergent Magnetic Resonance Imaging in Pregnant Patients with Suspected Appendicitis: A Single Center Perspective

PONE-D-23-00975R2

Dear Dr. Bufman,

We’re pleased to inform you that your manuscript has been judged scientifically suitable for publication and will be formally accepted for publication once it meets all outstanding technical requirements.

Kind regards,

Marc Reismann, MD, PhD

Academic Editor

PLOS ONE

Reviewers' comments:

Reviewer's Responses to Questions

**Comments to the Author**

1. If the authors have adequately addressed your comments raised in a previous round of review and you feel that this manuscript is now acceptable for publication, you may indicate that here to bypass the “Comments to the Author” section, enter your conflict of interest statement in the “Confidential to Editor” section, and submit your "Accept" recommendation.

Reviewer #2: All comments have been addressed

Reviewer #3: All comments have been addressed

2. Is the manuscript technically sound, and do the data support the conclusions?

Reviewer #2: Yes

Reviewer #3: Yes

3. Has the statistical analysis been performed appropriately and rigorously? 

Reviewer #2: N/A

Reviewer #3: Yes

4. Have the authors made all data underlying the findings in their manuscript fully available?

Reviewer #2: No

Reviewer #3: Yes

5. Is the manuscript presented in an intelligible fashion and written in standard English?

Reviewer #2: Yes

Reviewer #3: Yes

6. Review Comments to the Author

Reviewer #2: The manuscript has been satisfactory revised according to reviewer 3s comments. I have no further comments at this point.

Reviewer #3: (No Response)

7. PLOS authors have the option to publish the peer review history of their article (what does this mean?). If published, this will include your full peer review and any attached files.

Reviewer #2: **Yes: **Ellen Lokkegaard

Reviewer #3: **Yes: **Johannes Gossner

---

## [Editor Report · Acceptance letter]

3 Jul 2023

PONE-D-23-00975R2 

Findings on Emergent Magnetic Resonance Imaging in Pregnant Patients with Suspected Appendicitis: A Single Center Perspective 

Dear Dr. Bufman:

I'm pleased to inform you that your manuscript has been deemed suitable for publication in PLOS ONE. Congratulations! Your manuscript is now with our production department. 

Kind regards, 

on behalf of

Dr. Marc Reismann 

Academic Editor

PLOS ONE